# OpenReview forum: "Reward Shaping for (Inference-Time) Alignment: A Stackelberg Game Perspective"
_ICML.cc/2026/Conference — ICML 2026 regular_

### Official Review · Reviewer_LTR8 · 2026-03-10

**Soundness:** 3
**Presentation:** 3
**Significance:** 2
**Originality:** 2
**Overall Recommendation:** 3
**Confidence:** 3

**Summary:**

The main focus of this article is on how to achieve better alignment effects based on rewards when there is a KL regularization. Here, the author points out a common problem in the general process: even if the reward model is good, it is still possible that the rewards fail to maximize user utility due to the constraints that make the model approach the original version. The core idea lies in how to design the reward model more intelligently when it is known that the downstream model has such constraints.

The author of the article constructs this problem as a Stackelberg game and shows that the optimal reward has a threshold structure. Then, through Monte Carlo sampling, a practical approximate method is established. The author calls this method Stackelberg Reward Shaping (SRS), which combines existing Inference-time alignment methods, such as controlled decoding and ARGS. Experiments show that compared to the benchmark, its average reward and competitive winning rate continue to increase.

**Compliance With Llm Reviewing Policy:**

Affirmed.

**Final Justification:**

My concerns have only been partially addressed in the rebuttal.It remains unclear whether the theoretical results would collapse or become significantly more restrictive without the assumption of bounded rewards, and the sensitivity analysis regarding the implications of this assumption is insufficient. While the additional experiments help mitigate random noise, the issue concerning biased reward models still persists. Furthermore, it is not evident whether the proposed framework is indispensable for deriving the threshold structure, or if it mainly serves as a reformulation of the bi-level optimization problem.

Accordingly, my overall recommendation remains unchanged(3: Weak reject).

**Key Questions For Authors:**

1. How does the proposed method perform when the reward model contains significant noise or bias? Will threshold shaping amplify noise or bias?

2.In addition to Table 5 in the text, how stable is threshold estimation across different datasets and decoding policies? Can the author provide more quantitative analysis on the computational cost and performance improvement when increasing M?

**Limitations:**

Yes

**Strengths And Weaknesses:**

Strength:

1.The author proposes a logically coherent theoretical framework. The Stackelberg formula has been clearly defined, and the experimental and theoretical motivations are consistent.

2.The author formalizes the process of shaping rewards into a Stackelberg game. Although the concept of shaping rewards is not entirely new, considering the reward model as an optimization problem for a leader and a follower is a rather innovative perspective.

Weakness:

1.The author's theoretical derivation in this article has a clear logical structure, but it overly relies on several assumptions, such as bounded rewards and the accurate solving of KL-regularized objectives by followers. In fact, reward models may contain noise or bias.

2.Although the Stackelberg game theory is explained clearly, it is not entirely clear whether the Stackelberg framework is absolutely necessary for deriving threshold reward structures. The article does not analyze whether similar conclusions can be drawn even without using game theory formulas, or prove that this is not just conceptual.

3.Table 5 shows that the performance improvement on Eval-1 and ARGS is "slight", as the Monte Carlo sample size M increases. But this is only a partial verification, and the article does not analyze the complete experiment of computational cost vs. performance benefits. So is it worth the additional sampling cost?

4.This article only presents a conceptual Stackelberg diagram, but does not include a detailed end-to-end system flowchart, which lacks readability.

---

> ### Author Rebuttal · Authors · 2026-03-31
>
> We appreciate reviewers’ thoughtful questions. We address your concerns below and are happy to provide any clarification if needed.
>
> > ## **W1: Reliance on Assumptions.  Q1: Noisy Reward Model**
>
> **Rely on a bounded reward assumption?**
>
> The bounded reward is not an assumption but a design choice. We have shown empirically in Figure 2 and theoretically in Proposition D.4 that bounded reward helps mitigate reward hacking. We also cited related literature to support this design choice (Line 181).
>
> **Rely on accurately solving KL-regularized objectives?**
>
> This is a common assumption for theoretical analysis in alignment. See [3] as an example in inference-time alignment and Eq 4 in DPO [6] as an example for general alignment. The improved performance in Table 2 suggests that this assumption is reasonable for inference-time setting in practice.
>
> **Noisy Reward**
>
> Our method is robust to noise because bounded rewards prevent reward over-optimization, and the geometry of sigmoid transform is inherently noise-tolerant. For example, with threshold 3, the sigmoid suppresses small noise-induced differences (e.g., 0.9 vs. 1) while preserving meaningful gaps across the threshold (e.g., 1 vs. 5).  To evaluate robustness under noisy rewards, we apply a Gaussian noise to the reward, i.e., guide decoding with $r_U$ + N(0,1), while scoring responses using the ground-truth reward $r_U$​. The setup follows Eval1–2 in the main text.
>
> | Eval           | Vanilla Performance (Noisy Reward) | SRS Performance (Noisy Reward) | SRS Performance (Original Reward) |
> |----------------|-----------------------------------|--------------------------------|--------------------------|
> | Eval-1 (ARGS)  | 3.14                              | 3.31                           | 3.33                     |
> | Eval-2 (ARGS)  | 3.31                              | 3.40                           | 3.40                     |
>
> SRS consistently outperforms the vanilla methods in both settings. Moreover, its performance remains largely unchanged under Gaussian noise compared to the noiseless setting, indicating strong robustness.
>
> One can also use reward trained with robust optimization objectives [7]. Such a reward model penalizes noises.
>
> > ## **W2. Is game theory framework necessary or conceptual**
>
> The necessary component is the bilevel optimization where the upper level optimizes reward and lower level optimizes policy. In game theory terms, such bilevel optimization problem is referred to as a Stackelberg game. This formulation is natural for alignment, as the reward model serves as an incentive mechanism, and game theory provides a principled framework for optimizing incentives.
>
> > ## **W3:  Is the improvement of reward worth the additional sampling cost over increasing M Q2: More quantitative analysis on the computational cost and performance improvement when increasing M?**
>
> We clarify that in all main experiments, original methods (CD/ARGS) already need to sample multiple ($M$) responses, and we use the same $M$ across all baselines, so the comparison is fair. We use $M=10$ (the smaller value considered in the hyperparameter study), so SRS’s reward improvements reported in Table 2 are not due to increased computational cost.  Moreover, $M=10$ is both a practical sample size and the recommended choice in the underlying inference-time method [1].
>
> That said, we also provide a more detailed analysis of the effect of $M$, which shows $M = 10$ is sufficient and the benefit of more samples does not justify increased inference cost. We report average response time on 50 Eval-1 ARGS prompts.
>
> | Reward  | $M$  | Response Time (s) | Reward |
> |-------------|----|-------------------|--------|
> | Vanilla     | 10 | 29.51           | 3.23   |
> | Vanilla     | 20 | 54.70           | 3.20   |
> | SRS         | 10 | 29.75           | 3.33   |
> | SRS         | 20 | 55.68           | 3.40   |
>
> In Table 5, we already show that increasing M yields only a slight improvement in reward for SRS, and here we add the computation cost. Doubling $M$ roughly doubles the response time per prompt, since ARGS requires twice more reward model evaluations. This linear increase in inference cost is not justified by moderate reward gains, so we use $M=10$ in our main experiments. Also, the dominant computational cost comes from reward model evaluations (i.e., ARGS itself), while SRS introduces tiny overhead on top of base method.
>
> > ## **W4: Lack of System FlowChart**
>
> Thanks for the suggestion. **We will include the following system chart in the revised version**: https://github.com/icmlauthor/icml2026rebuttal/blob/main/system_flowchart.pdf
>
> ## **References:**
>
> References [1,3] are in our responses to Reviewer rFzk
>
> [6] Rafailov, Rafael, et al. "Direct preference optimization: Your language model is secretly a reward model." NeurIPS 2023.
>
> [7] Mandal, Debmalya, Paulius Sasnauskas, and Goran Radanovic. "Distributionally robust reinforcement learning with human feedback." arXiv (2025).

---

> > ### Author Rebuttal · Reviewer_LTR8 · 2026-04-01
> >
> > Thank you to the authors for the detailed rebuttal. The responses clarify several points and improve the presentation of the paper.
> >
> >
> > W1&Q1: The rebuttal gives useful clarification and adds evidence on noisy rewards. However, my concern is only partly resolved. It is still unclear whether the theory would break down or become much narrower without bounded rewards, and the sensitivity of the results to this assumption is not analyzed enough. The added experiment helps with random noise, but the issue of biased reward models remains.
> > W2: The authors explain why the Stackelberg formulation is a natural way to describe the problem. However, this does not fully address my concern. It is still unclear whether this framework is necessary for deriving the threshold structure, or whether it is mainly a reformulation of a bilevel optimization problem.
> > W3 & Q2: This response answers my question clearly. The added discussion of sampling cost and performance tradeoff largely resolves the concern.
> > W4: This is a clear clarification, and the promised system flowchart addresses the concern.
> >
> > Overall, the rebuttal addresses some of my questions, but the main theoretical concerns are only partly resolved. My overall recommendation remains unchanged (3: Weak reject).

---

> > > ### Author Response · Authors · 2026-04-08
> > >
> > > > # W1&Q1:
> > >
> > > ## Theory break without bounded reward?
> > >
> > > No, unbounded reward will not break our theory.
> > >
> > > Solving Program (3) [Line 169] with unbounded reward (i.e., set $B$ to infinity) leads the leader to assign infinite reward to the response maximizing $r_U$ (denoted $\tilde{y}$) and 0 to all others. The follower (LLM)’s best response is therefore to generate $\tilde{y}$ with probability 1. We clarify that this outcome is consistent with our framework and **DOES NOT**  break or narrow our theory. To see this, set $B \to \infty$, the threshold increases such that only $\tilde{y}$ exceeds it and receives reward $B$ (i.e., infinity), while all other responses receive 0. **The reward model still exhibits threshold structure.**
> > >
> > > ## Why bounded reward
> > >
> > > While unbounded rewards are consistent with theory, they are **UNDESIRABLE** in practice. Such shaping assigns a large reward only to $\tilde{y}$, causing the policy to collapse and generate only $\tilde{y}$. Besides loss of diversity, this is prone to reward hacking—since the reward model is trained on finite data, a low-quality $\tilde{y}$ may receive falsely high reward due to noise or bad OOD generalization. This effect is shown in Figure 2 (Appendix C.1). For vanilla ARGS (blue line), as reward strength 1/β increases from 0.5 to 1 (with 1 being a default choice in [3]), the average reward (solid line) increases, but the win rate against the base policy (dashed lines) decreases, and this trend persists at higher 1/β.
> > >
> > > Many alignment works, including DPO [6], argue that constraining the KL divergence between the aligned and base policies helps mitigate reward hacking, which occurs when the aligned policy deviates too far from the base policy. As noted in Line 185 and proven in Appendix D.4 (Prop1), we show that when rewards are bounded by B, the KL between aligned and base policy is bounded by B/β (up to constant). Thus, tightening the reward bound directly limits deviation from the base policy and reduces reward hacking. Empirically, Figure 2 (Appendix C.1) supports this: for any fixed 1/β, a tighter reward bound gets a higher win rate. With a sufficiently tight bound (purple line), reward hacking disappears, as the win rate remains stable across reward strengths.
> > >
> > > ## Biased reward
> > >
> > > **We add results on two New scenarios of biased reward**, one benign (moderate preference mismatch) and the other malicious (flipped preference).
> > >
> > > Case 1 (Benign): The user has a ground-truth preference but relies on a public reward model trained on others’ preference data, which is similar but not identical. To simulate this, we guide generation using one reward model while evaluating with another as ground truth. **New results (https://github.com/icmlauthor/icml2026rebuttal/blob/main/Benign.pdf)** show our method outperforms all baselines. This is because under moderate preference mismatch, our methods better capture user preference and maximize utility.
> > >
> > > To further show robustness, we evaluate cross-distribution generalization following Section 6.2 in [4]. We use a Q function learned on one dataset to guide generation on another dataset, where our method consistently outperforms. **See https://github.com/icmlauthor/icml2026rebuttal/blob/main/T.pdf**.
> > >
> > > Case 2 (Malicious): To simulate malicious reward, we use $r_U$ to score next token in ARGS and then invert the reward ordering, so low-quality outputs receive higher rewards. SRS still significantly outperforms the vanilla method, see **https://github.com/icmlauthor/icml2026rebuttal/blob/main/Malicious.pdf**. The intuition is SRS introduces a hyperparameter B; if reward model provider suspects a reward model is corrupted during validation, one can choose a small B to limit its impact.
> > >
> > > In validation, we use GPT judge to flag malicious rewards. In our case, the base policy achieves 76% win-tie rate against ARGS with malicious reward. No prior inference-time alignment work [1,2,3,4] considers malicious reward as the focus is optimizing a given preference; in practice, one should audit reward models before use, which has a separate literature.
> > >
> > > > ## W2: Is Stackelberg a reformulation & necessary
> > >
> > > Stackelberg game is not a reformulation of bilevel optimization—they are two names for the **Same Concept**. (see [lecture note](https://www.columbia.edu/~ck2945/files/ai_games_markets/lecture_note_16_stackelberg_games.pdf)). For any bilevel optimization problem, the upper-level problem can be seen as the leader’s problem, while the lower-level problem is the follower’s problem. We prove in Theorem 1 that the solution to our bilevel problem [Program 3] has a threshold structure.
> > >
> > > Since we already justified why game-theoretic framing is natural (the reward model acts as an incentive mechanism), and the reviewer acknowledges this, **we do not believe that using terms from different communities for the same concept becomes a weakness.**
> > >
> > > References:
> > > [1,2,3,4] are in our reply to Reviewer rFzk, [6] is in our reply to your prior questions.

---

### Official Review · Reviewer_imzK · 2026-03-10

**Soundness:** 2
**Presentation:** 3
**Significance:** 3
**Originality:** 3
**Overall Recommendation:** 4
**Confidence:** 3

**Summary:**

The paper studies the alignment problem with KL regularization. It observes that directly optimizing the regularized objective with the given reward function may fail to maximize the target. Thus, it motivates doing reward shaping. It gives the optimal choice as a threshold reward model and provides an efficient method to estimate the threshold with a Monte Carlo estimator. The proposed method is applied to inference-time generation to prove its effectiveness.

**Compliance With Llm Reviewing Policy:**

Affirmed.

**Final Justification:**

After the discussion, there remain some parts to be improved. However, the current paper is good enough as a borderline paper. Thus, I keep me score (4) and hold my opinion for acceptance.

**Key Questions For Authors:**

How does the proposed method perform on more general datasets? In particular, can the authors provide evidence of its effectiveness in training settings, as well as on benchmarks with verifiable rewards such as mathematics or coding tasks? I would consider increasing my score if the rebuttal demonstrates convincing improvements in these settings.

**Limitations:**

The paper discusses limitations in the impact statement.

**Strengths And Weaknesses:**

**Strength**:

1. The paper provides a clear formulation of the reward shaping problem using a Stackelberg game framework. Under this formulation, the authors derive rigorous theoretical results showing that the optimal reward shaping takes the form of a threshold function.

2. The paper proposes efficient methods for estimating the optimal reward shaping function.

3. Experiments are conducted to empirically demonstrate the effectiveness of the proposed approach.

**Weakness**:

1. In my understanding, the proposed reward shaping technique is not limited to inference-time alignment and could also be applied to training-based approaches such as RLVR. However, the paper does not discuss whether the method is effective in training settings or how it would perform within such frameworks.

2. In the theoretical analysis, the objective considered is purely reward maximization. However, in the experiments, additional metrics such as diversity and coherence are reported. It is therefore unclear why the proposed method would lead to improvements in these metrics, since they are not directly related to the optimization objective analyzed in the theory.

3. The abbreviation ARGS should be written out in full when it first appears in the paper.

4. The experimental evaluation is somewhat limited. I’m curious about the performance of the proposed algorithm on datasets with verifiable rewards, such as mathematics or coding benchmarks.

5. (Minor) The theoretical analysis focuses only on KL regularization. It would be helpful to discuss whether the framework can be extended to more general divergences, such as forward KL or other f-divergences.

---

> ### Author Rebuttal · Authors · 2026-03-31
>
> We thank the reviewer for their thoughtful comments. We address your concerns below and are happy to provide further clarification if needed. Thanks!
>
> > ## **W1: Is Proposed method not limited to inference-time alignment, is it effective in RLVR? W4: No experiments on math or coding benchmark. Q1: More general datasets, such as coding and math**
>
> **Why not test on Math and Coding Benchmark:**
>
> Our method is designed for alignment, and it’s not designed for reasoning tasks with verified rewards. The central challenge that our paper is trying to address is the conflict between user preferences ($r_U$) and the model’s inherent preferences. But there is no user preference in reasoning tasks, because reasoning tasks have objective answers. Evaluation on reasoning tasks such as mathematics or coding is less aligned with the primary focus of our work, as these domains rely on verifiable correctness rather than preference-based rewards.
>
> Moreover, we note that RLVR rewards already exhibit an implicit threshold structure, where the reward is 1 only when the answer is correct and 0 otherwise, leaving little room for improvement through threshold reward shaping via our method.
>
> **Why not focus on train-time:**
>
> Our work focuses on inference-time alignment, a well-established and broad enough research topic with a substantial body of prior works in venues such as ICML, NeurIPS, and ICLR [1,2,3,4]. Consistent with prior work, we test on alignment benchmarks including HH-RLHF and SHP [1,3,4].
>
> Extending our method to the train-time setting is an interesting direction but not immediate. Inference-time formulation has a fixed follower (base policy), whereas in train-time the policy evolves during optimization. Hence, the analysis for train-time setting brings in extra theoretical concerns like learning dynamics. For example, [5] shows that an accurate reward model may fail in RLHF if the reward variance is not large enough to yield a strong gradient signal. Such considerations are absent in inference-time settings, where no gradient-based optimization is performed. Moreover, empirically, our hyperparameters $(B, \alpha)$ are chosen with respect to a static follower, and it needs further investigation to understand how to select hyperparameters when the follower is adaptive. For these reasons, extending our framework to train-time setting requires much new analysis of training dynamics and stability, and we leave a comprehensive analysis of train-time setting as an important direction for future work.
>
> > ## **W2: Why proposed method lead to improvement in diversity and coherence?**
>
> We did not claim an improvement in diversity and coherence as the difference between methods is small in Table 2. We only report them to "maintain comparable levels" compared to baseline performance (See line 362 on the right column).
>
> Prior literatures [1,3,4] also only aim to maximize reward in their objectives, but they all report diversity and coherence because these metrics are useful for detecting reward hacking as found in previous literature (see Sec 6.3 in [4]). Although the primary objective is reward maximization, it is important to ensure that gains in reward do not come at the expense of decreased diversity and coherence. The fact that our method maintains the same level of diversity and coherence as the Vanilla policy is desirable, as it it means our method improves reward without hurting diversity or coherence.
>
> > ## **W3: The abbreviation ARGS should be written out in full when it first appears in the paper.**
>
> Thank you for pointing that out. We will update our writing in the revised version.
>
> > ## **W5: Can the framework be extended to general divergences?**
>
> The Stackelberg framework is independent of the divergence used, and we focus on KL divergence because it’s most widely used in alignment and provides closed-form solutions. It’s interesting to generalize our theoretical analysis to f-divergence as future work. Thank you for the suggestion.
>
>
> **References:**
>
> References [1,2,3,4] are in our responses to Reviewer rFzk.
>
> [5] Razin, Noam, et al. "What makes a reward model a good teacher? an optimization perspective." NeurIPS 2025.

---

> > ### Author Rebuttal · Reviewer_imzK · 2026-04-01
> >
> > Thanks for your detailed response. It addresses part of my questions. However, I still find the explanation regarding my main question, specifically, why the method is not evaluated on mathematical or other verifiable-reward tasks unconvincing.
> >
> > The authors state that such benchmarks are not considered because the answers are objective and do not involve user preferences. However, this reasoning seems inconsistent with the formulation presented in the paper. While the reward function is described as being learned from user preferences, the framework itself does not appear to impose any structural assumptions on how this reward is obtained. At a theoretical level, the analysis only requires a reward function $r_U$. In mathematical tasks, for example, there are numerous works on training rewards functions (outcome-based or process-based). These can be viewed as valid instantiations of $r_U$. You can apply reward shaping to these trained reward functions, and use the verifiable rewards for performance evaluation. From this perspective, these settings seem fully compatible with the framework. Therefore, I respectfully disagree with the claim that the proposed method is not designed for mathematical or other verifiable-reward tasks.
> >
> > Due to these concerns, I will maintain my score at 4, while still holding an overall positive opinion of the work.

---

> > > ### Author Response · Authors · 2026-04-08
> > >
> > > We thank the reviewer for the feedback. We agree that in the mathematical framework, $r_U$​ does not necessarily have to represent user preference. However, we believe there is some subtle difference between the alignment tasks and reasoning tasks which makes the direct adaptation of our method to reasoning tasks non-immediate.
> > >
> > > In our framework, $r_U$ corresponds to the evaluation reward, as the objective is to optimize performance under the final evaluation metric. However, if we understand correctly, the reviewer’s proposed pipeline instead uses process-based rewards as $r_U$, rather than the final evaluation model. This change would alter the current formulation of our framework. On the other hand, if one directly uses a 0/1 evaluation reward as $r_U$, it already exhibits a threshold structure, and hence additional threshold-based reward shaping is unlikely to provide benefit. That said, we appreciate the reviewer’s insightful suggestion. We believe it is an interesting direction for future work to extend our framework to reasoning tasks by incorporating process-based rewards, and we thank the reviewer for highlighting this opportunity. **We will include a discussion of this potential extension to reasoning tasks in the revised version of the paper.**
> > >
> > > Meanwhile, we also respectfully note that, while our method could possibly be extended to other settings, it is challenging for a single conference paper to cover all possible applications. Our method is naturally inspired by alignment tasks, and we focus on a scope consistent with prior ICML/NeurIPS/ICLR works [1,2,3,4] and provide a comparable level of empirical evaluation (as discussed in our initial rebuttal). We hope this scope is recognized, and that extending to reasoning tasks is viewed as an interesting possible direction for future work rather than a limitation of the current paper.
> > >
> > >
> > > **References**:
> > > References [1,2,3,4] are in our responses to Reviewer rFzk.

---

### Official Review · Reviewer_pcbc · 2026-03-13

**Soundness:** 3
**Presentation:** 3
**Significance:** 2
**Originality:** 3
**Overall Recommendation:** 4
**Confidence:** 3

**Summary:**

The authors assess a fundamental issue in KL-regularized alignment: directly optimizing a learned reward model may be suboptimal for maximizing user utility because the KL term preserves biases from the base model. The article's fundamental theme concerns how to optimally shape the reward used for alignment rather than how to optimize the policy given a fixed reward. To address this, the paper formulates reward design as a Stackelberg game, derives an optimal threshold-structured reward under a bounded-reward constraint, and proposes a practical soft relaxation called Stackelberg Reward Shaping (SRS) that can be approximated with Monte Carlo samples from the base model. The method is then integrated into two inference-time alignment methods, CD and ARGS, and evaluated on HH-RLHF and SHP using Qwen3-8B and Llama3-8B-Instruct backbones. The reported results show consistent gains in average reward and around 66% average GPT-4 win–tie rate over baselines.

**Compliance With Llm Reviewing Policy:**

Affirmed.

**Final Justification:**

Thank you for the detailed response. While I still have some reservations regarding the evaluation of model scaling, I appreciate the clarifications provided and the additional context offered by the authors. Overall, I believe the paper now meets the bar for acceptance, albeit marginally, and I have accordingly increased my score.

**Key Questions For Authors:**

N/A

**Limitations:**

Yes

**Strengths And Weaknesses:**

Strengths:
* The paper studies an important and somewhat overlooked question. Most prior work treats the reward model as fixed, whereas this paper asks whether the reward itself should be reshaped under KL regularization. This is a meaningful perspective shift, and the motivating example clearly illustrates why direct reward optimization may be misaligned with true user utility.
* The theoretical development is reasonably clean. The Stackelberg formulation is intuitive, the optimal reward is shown to have a threshold structure, and the paper further derives a practical Monte Carlo approximation plus a soft relaxation to improve robustness.
* The method is lightweight and practically motivated. It is designed to plug into existing inference-time alignment methods, and the reported overhead for SRS-ARGS is very small. The main trend that SRS improves average reward over vanilla and heuristic shaping baselines is consistent across the reported settings.

weaknesses:
* Is the theoretical approach scalable? It would be helpful to see more extensive experiments on larger models and across different model families.
* The evaluation remains overly tied to the same reward model family used as the optimization target. Average reward is computed by r_U, which is instantiated using public Skywork reward models, and GPT-4 judgment is the only complementary quality check. This leaves open the usual concern that gains are partly evaluator-specific. A stronger paper would include either human evaluation, cross-reward-model evaluation, or transfer tests showing that SRS improves outputs under alternative judges not used in shaping
* The usefulness of the Diversity and Coherence metrics is unclear. The results show little difference between methods on these metrics, which raises questions about their effectiveness in distinguishing model performance.
* The comparison to prior reward-shaping methods is limited. The paper argues that train-time shaping methods are not directly transferable to inference-time alignment, which is reasonable, but the actual baselines are mainly Minmax and Meanstd heuristics derived from one prior work. More diverse baselines, especially stronger inference-time methods, would make the empirical case more convincing.

---

> ### Author Rebuttal · Authors · 2026-03-31
>
> We thank the reviewer for their thoughtful comments. We address your concerns below and are happy to provide further clarification if needed. Thanks!
>
> > ## **W1: Is … scalable? … experiments on larger model families?**
>
> We would like to note that, as a small academic lab operating on a shared university cluster, running extensive experiments across large model families is resource-intensive. **The GPU cluster was temporarily down during the rebuttal period, so we have to prioritize other less resource intensive tasks**.
>
> Moreover, prior major inference-time alignment works published in ICML/NeurIPS/ICLR [1,2,3,4] all evaluate methods on models in the 7B–8B range or smaller. We hope the reviewer will take these practical constraints into consideration.
>
> > ## **W2: Cross-reward evaluation or transfer test**
>
> We thank the reviewer for this suggestion. We perform the following three experiments to indicate the robustness of our method under cross-reward/transfer test settings. **We will include these results and discussions in the revised version of our work.**
>
> In **Experiment 1**, we use one reward model to guide decoding (“Guide Reward”) and a different reward model for evaluation (“Evaluation Reward”). We include one result with UltraRM-13B, which is outside the Skywork reward family. SRS remains robust, outperforming baseline reward shaping methods under both settings.
>
> | Eval           | Guide Reward   | Evaluation Reward | Vanilla | Minmax | Meanstd | SRS   |
> |----------------|----------------|-------------------|---------|--------|---------|-------|
> | Eval-1 (ARGS)  | Skywork-Qwen   | UltraRM-13B       | -4.58   | -4.60  | -4.60   | -4.52 |
> | Eval-3 (ARGS)  | Skywork-Llama  | Skywork-Qwen      | 3.20    | 3.20   | 3.09    | 3.40  |
>
> In **Experiment 2**, we apply a noisy reward to guide inference-time decoding while evaluating using the original reward, also making the optimization target different from the evaluation reward. Specifically, we apply a Gaussian noise to the reward, i.e., guide decoding with $r_U$ + N(0,1), while scoring responses using the ground-truth reward $r_U$​. The setup follows Eval1–2 in the main text.
>
> | Eval           | Vanilla Performance (Noisy Reward) | SRS Performance (Noisy Reward) | SRS Performance (Original Reward) |
> |----------------|-----------------------------------|--------------------------------|--------------------------|
> | Eval-1 (ARGS)  | 3.14                              | 3.31                           | 3.33                     |
> | Eval-2 (ARGS)  | 3.31                              | 3.40                           | 3.40                     |
>
> SRS consistently outperforms the vanilla policy in both settings, and its performance remains largely unaffected by Gaussian noise in the reward, indicating strong robustness against noises.
>
> In **Experiment 3**, we follow the same transfer test as [4] (See its Section 6.2). We use the learned Q function $\hat{Q}$, trained on Eval-2 (CD) with the SHP dataset, to prompts in HH-RLHF dataset to test the robustness of different methods across prompt distribution. Our method outperforms all baselines. This shows our method robustly generalizes to input distribution different from that where the value function is trained on.
>
> | Method   | Reward |
> |----------|------------|
> | Vanilla  | 3.16       |
> | Minmax   | 2.84       |
> | Meanstd  | 2.96       |
> | SRS      | 3.28       |
>
> > ##  **W3: Usefulness of diversity and coherence unclear**
>
> Reporting diversity and coherence is standard in the inference-time alignment literature  [1,3,4]. Diversity and Coherence are useful metrics for detecting reward hacking as found in previous literature (see Sec 6.3 in [4]).
>
> Although the primary objective is reward maximization, it is important to ensure that gains in reward do not come at the expense of diversity and coherence; thus, they serve as sanity checks. “Little difference between methods on these metrics” is desirable because it means our method improves reward without hurting diversity or coherence.
>
> > ##  **W4: More diverse baseline**
>
> We are not proposing new inference-time decoding methods, and we are proposing a new reward shaping method that can be plugged into existing inference-time decoding methods. To the best of our knowledge, we are not aware of other reward shaping baselines that are designed for this purpose.
>
> ## References:
>
> References [1,2,3,4] are in our responses to Reviewer rFzk.

---

> > ### Author Rebuttal · Reviewer_pcbc · 2026-04-03
> >
> > I appreciate the authors’ rebuttal and additional experiments. Some concerns are addressed, but key issues remain only partially resolved.
> >
> > **Comment 1 (Scale):** Unresolved but understandable. The resource explanation is reasonable, but it does not provide evidence on generalization to larger model families.
> >
> > **Comment 2 (Evaluator dependence):** Partially resolved. The additional cross-reward, noisy reward, and transfer experiments are helpful and directly respond to the concern. However, the gains under alternative reward models (e.g., UltraRM-13B) appear relatively small, which raises questions about the practical significance and robustness of the improvements beyond the original reward family.
> > **Comment 3:** Resolved.
> >
> > **Comment 4:** Mostly resolved, but some uncertainty remains regarding relation to similar methods.
> >
> > Overall, the rebuttal is helpful, but core concerns about robustness and empirical support are not fully resolved. My overall recommendation remains unchanged (3: Weak reject).

---

> > > ### Author Response · Authors · 2026-04-08
> > >
> > > Thanks!
> > >
> > > > ## Comment 1
> > >
> > > Thanks for recognizing our resource constraints. We clarify our effort and the practical challenges of working with large models under limited rebuttal time.
> > >
> > > We tried to include results for Gemma3-27B during rebuttal. However, ARGS requires the reward model to share the backbone tokenizer. Without a compatible public reward, we must fine-tune one from Gemma, which is time- and GPU- consuming even with LoRA. For CD, we must generate a new offline dataset using Gemma to train the Q function. With a larger backbone, both offline and inference-time generation are more costly. **This is not a limitation of our method**: these costs stem from the base inference-time method, while our method only adds a lightweight root-finding step independent of model size which is very fast (Section 7.3).
> > >
> > > Given crowded cluster and large experiment demands during rebuttal, we have to prioritize other lighter tasks. **We plan to include results and further analysis on Gemma3-27B in the revised version.**
> > >
> > > > ## Comment 2 (Evaluator Dependence)
> > >
> > > Denote UltraRM as Ultra, Skywork-Qwen as S-Q and Skywork-LLama as S-L.
> > >
> > > **New Experiment with Larger Gains**
> > >
> > > We introduce a new cross-reward evaluation using Ultra and S-L, and show in the next section that it's more representative than our prior rebuttal due to reduced preference mismatch between the guide and evaluation rewards. In the updated experiment (**See https://github.com/icmlauthor/icml2026rebuttal/blob/main/cross.pdf**), our method remains robust, outperforming baselines and achieving larger gains than previously reported.
> > >
> > > **Why prior gain is moderate under Ultra**
> > >
> > > We argue that the moderate performance gain in first-round rebuttal under Ultra is primarily driven by preference mismatch between Ultra and S-Q, rather than a lack of robustness of our method. Our updated result is more representative under cross-reward with similar preferences. Since reward models are trained on different datasets and annotator preferences, there is inherent disagreement between them. We find the updated setting (Ultra vs. S-L) has a much smaller preference mismatch than prior setting (Ultra vs. S-Q), which explains the larger gain.
> > >
> > > We measure disagreement among reward models using 3 metrics. We ask the base Qwen3 model to generate 10 responses per prompt in Eval-1, and rank them using Ultra, S-Q, and S-L. This gives 3 rankings per prompt (45 pairwise comparisons under each ranking). We then compute (1) the Kendall tau distance, the number of pairwise disagreements, averaged over 300 prompts, (2) fraction of pairwise comparisons on which two reward models disagree, (3) fraction of prompts where reward models agree on the top response. **Result in https://github.com/icmlauthor/icml2026rebuttal/blob/main/Mismatch.pdf**
> > >
> > > Besides smaller Kendall tau distance, Ultra agrees with S-L twice as often as with S-Q on which response is the best. The disagreement rate between Ultra and S-Q is about 10% higher than that between Ultra and S-L, indicating greater preference inconsistency. When the guide reward (S-Q) pushes the policy to a more distinct preference, it’s expected that it has smaller gain under evaluation reward (Ultra).
> > >
> > > The cross reward test is meaningful when both reward models capture similar preferences. Our new experiment shows that under this condition, our method generalize to alternative reward model reliably.
> > >
> > > > ## Comment 4
> > >
> > > To the best of our knowledge, no prior work views alignment reward shaping from a game theory angle. In Line 342, we note that LSC [8] and PAR [9] are not directly transferable from train time. We discuss here.
> > >
> > > LSC: $r = \log \sigma(r-r_{ref}^{.85})$, where $r_{ref}^{.85}$ is the 85th percentile of Gaussian fitted to the reference reward distribution (using its mean and variance).
> > >
> > > PAR: $r = 1/M \sum_{m=1}^M \sigma(r-r_{ref}^m)$, where r is the reward of the trained policy, and $r_{ref}^{m}$ is reward of m-th reference response.
> > >
> > > Both methods use reward of a reference (base) policy to regulate reward of trained policy (r), with the sigmoid form derived from the Bradley–Terry model and interpreted as the probability that a response outperforms a reference. These formulations require comparisons between two policies, which are unavailable at inference time where only the base policy can be sampled.
> > >
> > > We adapt both methods to inference time by letting r and $r_{ref}$ both be rewards of base policy samples, and report **New experiment results in https://github.com/icmlauthor/icml2026rebuttal/blob/main/Baseline.pdf** for these transformed baselines. Since r and $r_{ref}$ are now rewards of the same policy, this breaks the intended cross-policy pairwise comparison, leading these methods to underperform existing baselines. **We will add the new results in the revised version.**
> > >
> > > References:
> > >
> > > [8] Transforming and combining rewards for aligning large language models. ICML 2024
> > >
> > > [9] Reward shaping to mitigate reward hacking in rlhf. Arxiv 2025

---

### Official Review · Reviewer_rFzk · 2026-03-14

**Soundness:** 3
**Presentation:** 3
**Significance:** 3
**Originality:** 3
**Overall Recommendation:** 5
**Confidence:** 3

**Summary:**

In this paper, the authors study the problem of inference-time alignment for LLMs. The authors identify a critical flaw in standard alignment pipelines: directly optimizing a learned reward model under a KL-divergence constraint often fails to maximize true user utility, particularly when the base policy exhibits strong biases. To address this, the authors formulate reward model optimization as a Stackelberg game. In this, the reward model provider (the leader) designs a shaped reward, and the LLM (the follower) best responds by generating a KL-regularized policy. Theoretically, the authors prove that the optimal reward model follows a distinct threshold structure. Practically, they introduce Stackelberg Reward Shaping (SRS), which approximates this optimal threshold using Monte Carlo sampling and applies a soft sigmoid relaxation for robustness. When integrated into inference-time methods such as Controlled Decoding (CD) and ARGS, SRS consistently improves average reward while maintaining generation diversity and coherence, achieving a win-tie rate of over 66% against baselines in GPT-4 evaluations.

**Compliance With Llm Reviewing Policy:**

Affirmed.

**Final Justification:**

The new results support the robustness of SRS to hyperparameter configuration. I also appreciate the clarification on the thresholding mechanism. As most of my concerns have been addressed, I now hold a more positive view of this work and have increased my score accordingly.

**Key Questions For Authors:**

1. How sensitive is the optimal (B, α) configuration to changes in the base model or reward model?

**Limitations:**

Yes

**Strengths And Weaknesses:**

> **Strengths**

1. The formulation of the reward model optimization as a Stackelberg game is interesting and well-motivated. By casting the reward model provider as a Stackelberg leader and the KL-regularized LLM as the follower, the authors derive a closed-form characterization of the optimal reward that principally counteracts base policy biases.

2. The SRS framework calculates the reward threshold via bisection on Monte Carlo samples, as a result, introducing minimal latency for inference pipelines.

3. SRS achieves the highest average reward across all four evaluation settings while maintaining diversity and coherence comparable to both the base policy and all baselines

> **Weaknesses**

1. Based on Figure 4, performance seems to be dependent on the reward bound $B$ and shaping strength $\alpha$. The ablation study shows that using a large $B$ with a hard threshold ($\alpha \rightarrow \infty$) creates an imbalanced incentive structure that degrades user utility.

2. The experiments currently use only 8B-parameter models on two alignment benchmarks (HH-RLHF and SHP), both of which test helpfulness/harmlessness. It would be interesting to see results on larger, stronger models, and more diverse alignment setups.

---

> ### Author Rebuttal · Authors · 2026-03-31
>
> We thank the reviewer for their thoughtful comments. We address your concerns below and are happy to provide further clarification if needed. Thanks!
>
> > ## **Q1: How sensitive is the optimal (B, α) configuration**
>
> We conduct the following three robustness analyses for parameters (B, α).
>
> In **Experiment 1**, we fix the eval and take the optimal parameter from one underlying method (Source Setting) to another method (New Setting). The result either matches or exceeds the best baseline (best among Vanilla, Minmax, Meanstd), indicating robustness of the hyperparameter.
>
> | Source Setting | Src Optimal (B, α) | New Setting | Performance under New Method | Best Baseline Performance |
> |------------|--------------------|---------|----------------------------|-----------------------------|
> | Eval-1 (CD)   | (12, 1.5) | Eval-1 (ARGS) | 3.24 | 3.24 |
> | Eval-2 (CD) | (10, 2.0) | Eval-2 (ARGS) | 3.40 | 3.26 |
>
> In **Experiment 2**, we perturb the reported optimal parameter and see whether the performance remains strong. This finding shows that SRS’s performance is stable around optimal parameters.
>
> | Settings  | Optimal (B, α) | Perturbed (B, α) | Reward under Perturbation | Best Baseline Reward |
> |----------------|----------------|------------------|--------------------------------|---------------|
> | Eval-3 (ARGS)  | (15.0, 1.0)    | (15.0, 1.5)      | 1.94                           | 1.87          |
> | Eval-3 (ARGS)  | (15.0, 1.0)    | (12.0, 2.5)      | 2.04                           | 1.87          |
> | Eval-2 (CD)    | (10, 2.0)      | (10, 3.0)        | 3.23                           | 3.10          |
> | Eval-2 (CD)    | (10, 2.0)      | (10, 1.0)        | 3.16                           | 3.16          |
>
> In **Experiment 3**, following Section 6.2 in [4], we evaluate cross-distribution robustness by applying the learned Q function $\hat{Q}$, trained on Eval-2 (CD) with the SHP dataset, to Eval-1 (CD) on HH-RLHF prompts. The following table shows $\hat{Q}^{SRS}$, trained with optimal $(B, \alpha)$ configuration from Eval-2, generalizes robustly to a different input distribution in Eval-1.
>
> | Method   | Performance |
> |----------|------------|
> | Vanilla  | 3.16       |
> | Minmax   | 2.84       |
> | Meanstd  | 2.96       |
> | SRS      | 3.28       |
>
> > ## **W1: Based on Figure 4, performance seems to be dependent on the reward bound B and shaping strength. The ablation study shows that using a large B with a hard threshold creates an imbalanced incentive structure that degrades user utility.**
>
> There seems to be some misunderstanding of our method in the reviewer’s comment.
>
> (1) First, our method doesn’t apply a hard threshold. We specifically choose soft threshold to improve robustness, and we include one failure example for the hard threshold in the ablation study to justify our soft threshold design, so a critique on hard threshold should not be seen as a weakness of our method.
>
> (2) $(B, \alpha)$ are hyperparameters, and it’s very common for ML algorithms’ performance to depend on its hyperparameter.
> It is common in the inference time alignment literature to choose a small hyperparameter to prevent reward hacking: For example, see Section 6.3 in [4], where they recommend a medium step size (i.e., $\alpha$) for this purpose.
>
> (3) Moreover, the issue highlighted by the reviewer is not unique to our method, but rather arises from using a large reward bound B. We show empirically in Figure 2 and theoretically in Proposition D.4 that large reward bound may cause reward hacking.
>
> That said, in response to Q1 (Experiment 2) we conducted an analysis showing that the optimal parameters $(B, \alpha)$ are robust and stable within a neighborhood. As a result, a simple hyperparameter sweep is sufficient to identify this region and recover good performance, making hyperparameters relatively easy to tune in practice.
>
> > ## **W2.  The experiments currently use only 8B-parameter models...**
>
> We would like to note that, as a small academic lab operating on a shared university cluster, running extensive experiments across large model families is resource-intensive. **The GPU cluster was temporarily down during the rebuttal period, so we have to prioritize other less resource intensive tasks**.
>
> Moreover, prior major inference-time alignment works published in ICML/NeurIPS/ICLR [1,2,3,4] all evaluate methods on models in the 7B–8B range or smaller. We hope the reviewer will take these practical constraints into consideration.
>
> ## References:
>
> [1] Khanov, Maxim, Jirayu Burapacheep, and Yixuan Li. "Args: Alignment as reward-guided search." ICLR 2024.
>
> [2] Mudgal, Sidharth, et al. "Controlled decoding from language models." ICML 2024.
>
> [3] Chakraborty, Souradip, et al. "Transfer q-star: Principled decoding for llm alignment." NeurIPS 2024.
>
> [4] Kong, Lingkai, et al. "Aligning large language models with representation editing: A control perspective." NeurIPS 2024.

---

> > ### Author Rebuttal · Reviewer_rFzk · 2026-04-03
> >
> > Thank you for the additional experiments. The new results support the robustness of SRS to hyperparameter configuration. I also appreciate the clarification on the thresholding mechanism. As most of my concerns have been addressed, I now hold a more positive view of this work and have increased my score accordingly.

---

> > > ### Author Response · Authors · 2026-04-08
> > >
> > > We thank the reviewer for the helpful feedback and are glad the concerns are addressed. We appreciate the discussion.

---

### Decision · Program_Chairs · 2026-04-30

**Decision:**

Accept (regular)

**Comment:**

The paper argues that standard inference-time LLM alignment is suboptimal because KL regularization causes the aligned model to inherit biases of the base policy that conflict with user preferences, and proposes a reward reshaping scheme that counteracts this bias and can be integrated into existing inference-time alignment methods. The analysis is based on modeling the reward designer and the KL-regularized LLM as two players in a Stackelberg game.

Three out of four reviewers recommended acceptance, finding the problem formulation interesting and the proposed method well-motivated. In the revised version, the authors should address all reviewer comments, incorporating the additional arguments and results provided during the rebuttal. From my perspective, the most consistent concern is that the evaluation relies on reward models from the same family used during optimization, making it unclear whether gains reflect genuine improvements in user utility or are partly specific to the reward model. The cross-reward, noisy reward, and biased reward experiments provided during the rebuttal should be incorporated into the revised paper. It would also strengthen the paper to include results on larger model families, as indicated by the authors during the rebuttal.